# Mercury and Arsenic Discharge from Circumneutral Waters Associated with the Former Mining Area of Abbadia San Salvatore (Tuscany, Central Italy)

**DOI:** 10.3390/ijerph19095131

**Published:** 2022-04-23

**Authors:** Marta Lazzaroni, Marino Vetuschi Zuccolini, Barbara Nisi, Jacopo Cabassi, Stefano Caliro, Daniele Rappuoli, Orlando Vaselli

**Affiliations:** 1Department of Earth Sciences, University of Florence, Via G. La Pira 4, 50121 Florence, Italy; 2INSTM, National Interuniversity Consortium of Materials Science and Technology, Via G. Giusti 9, 50121 Florence, Italy; 3CNR-IGG, Institute of Geosciences and Earth Resources, Via G. La Pira 4, 50121 Florence, Italy; barbara.nisi@igg.cnr.it (B.N.); jacopo.cabassi@igg.cnr.it (J.C.); 4Department of Earth, Environmental and Life Sciences, Corso Europa 26, 16132 Genoa, Italy; marino.zuccolini@unige.it; 5INGV—Istituto Nazionale di Geofisica e Vulcanologia, Via Diocleziano 328, 80124 Napoli, Italy; stefano.caliro@ingv.it; 6Unione dei Comuni Amiata-Val d’Orcia, Unità di Bonifica, Via Grossetana 209, Piancastagnaio, 53025 Siena, Italy; d.rappuoli@uc-amiatavaldorcia.si.it; 7Parco Museo Minerario di Abbadia San Salvatore, Via Suor Gemma 1, Abbadia San Salvatore, 53021 Siena, Italy

**Keywords:** abandoned Hg mines, Mt. Amiata, river chemistry, geoaccumulation index, total mass load, mercury, arsenic

## Abstract

Dissolved and suspended toxic elements in water discharged from abandoned and active mining areas pose several critical issues, since they represent a threat to the environment. In this work, we investigated the water, suspended particulates, and stream sediments of a 2.1 km long creek (Fosso della Chiusa) that is fed by waters draining the galleries of the abandoned Hg mining area of Abbadia San Salvatore (Mt. Amiata, Tuscany, central Italy). The geochemical results show evidence that the studied matrices are characterized by relatively high concentrations of Hg and As, whereas those of Sb are generally close to or below the instrumental detection limit. Independent of the matrices, the concentration of As decreases from the emergence point to the confluence with the Pagliola creek. In contrast, Hg concentrations display more complex behavior, as water and sediment are mainly characterized by concentrations that significantly increase along the water course. According to the geoaccumulation index (I_geo_), sediments belong to Class 6 (extremely contaminated) for Hg. The I_geo_ of As varies from Class 6, close to the emergence, to Class 2 (moderately contaminated), dropping to Class 0 (uncontaminated) at the confluence with the Pagliola creek. Finally, the total mass load of Hg and As entering the Pagliola creek was computed to be 1.3 and 0.5 kg/year, respectively, when a mean flow rate of 40 L/s was considered. The calculated loads are relatively low, but, when the Fosso della Chiusa drainage basin is taken into account, the specific load is comparable to, or even higher than, those of other mining areas.

## 1. Introduction

The Mt. Amiata area (1738 m above sea level, southern Tuscany, central Italy) is dominated by a homonymous volcanic system (0.3–0.2 Ma old), and is well-known for the exploitation of geothermal resources to produce electricity (e.g., [1]). In addition, this relatively small sector of Tuscany belongs to the so-called Mediterranean Hg Belt (e.g., [2] and references therein) and hosts one of the most important sites worldwide for the exploitation of cinnabar (HgS)-rich ore deposits [3,4], from which liquid mercury is produced. The Mt. Amiata Hg-district, a *“World class Ore Deposit*”, according to its size, profitability and tenor [5], pertains to the Tuscan metallogenic province, named “Colline Metallifere”, which includes numerous metallic poly-sulfide ores that have been exploited since the Etruscan times, e.g., [3,6,7,8,9]. At least seven HgS-rich ore deposits around the Mt. Amiata area were cultivated up to the middle 1970s, and that of Abbadia San Salvatore, located in the eastern flank of the volcano, was by far the most important. Here, the mercury mine operated underground and consisted of about 35 km of galleries delving down to a depth of 400 m below ground.

The mined ore deposits were mainly found in: (i) *hydrothermal breccias pipes* and (ii) *conformable hydrothermal breccia bodies* [4,10,11]. The first hosted HgS mineralization inside an argillitic matrix (mostly kaolinite and montmorillonite) cross-cutting the Mt. Amiata volcanics, while the second was predominantly found at the contact point between sedimentary and volcanic sequences [4,12]. Once mining activity definitively shut down (early 1980s), it was decided to close all the galleries’ entrances with concrete due to safety reasons [13]. Accordingly, no access to the mining galleries is presently available. The meteoric water interacts underground with *hydrothermal breccias pipes* and *bodies,* and discharges outside the mining area following two main paths: (i) Galleria Nuova Italia or “Livello Italia” (hereafter Galleria Italia), situated at 800 m a.s.l. [10,14,15], and (ii) Galleria Ribasso, the latter being located at about 400 m a.s.l. [14]. The infiltrating water drains different mine sectors at different flow rates, as follows: (a) qz-latite volcanics, ca. 8 L/s; (b) trachytic rocks, ca. 15 L/s; (c) “Pozzo Trivella”, a mine well located north of the mining area with reddish colored water, due to the relatively high concentrations of Fe-oxy-hydroxides, is observed, ca. 40 L/s; (d) volcanic and sedimentary formations, ca. 3–4 L/s. Consequently, about 67 L/s of water is expected to be discharged at the surface. Despite the fact that the flow rate at Galleria Italia may have pulses up 100 L/s during intense precipitations, the mean value is relatively stable at about 40 L/s [15]. The remaining amount of water, not released from Galleria Italia, should be discharged at Galleria Ribasso. Nevertheless, for roughly two decades this site has remained completely dry. Due to roof and wall collapses in the mine tunnel, the water was likely diverted and now follows unknown paths [14].

The Galleria Italia waters are slightly acidic (pH ≈ 6) and Ca-SO_4_ in composition, with TDS (Total Dissolved Solids) of about 1000 mg/L. They have relatively high concentrations of Fe, Al, Mn, As, and dissolved CO_2_, but a low concentration of Hg [15]. Owing to the physicochemical features of Galleria Italia waters, they can be classified as “Non-acidic Metalliferous Mine Drainage (MMD)” or “Circumneutral Mine Drainage (CMD)”, e.g., [16,17,18]; i.e., they are not acidic, as is commonly observed when polymetallic sulfide-rich deposits interact with meteoric waters ([17] and reference therein). The Galleria Italia waters feed a 2.1 km long creek, named Fosso della Chiusa, that eventually enters the Pagliola creek (Figure 1). Previous studies [18] provided a geochemical characterization of the water discharging from Galleria Italia; in this work, we present and discuss the geochemical and isotopic (oxygen and hydrogen in H_2_O and sulfur in SO_4_) composition of these surface waters, from the spring to the mouth, in order to: (i) characterize the distribution of Hg, As, and Sb in the liquid phase, suspended solids, and stream sediments; (ii) estimate how As, Hg, and Sb impact sediments by using the geoaccumulation index (I_geo_ [19]), and, finally, (iii) compute the yearly loads of the chalcophile elements that enter the surface riverine network from Fosso della Chiusa. The yearly loads were then compared with those of other water courses draining mining areas worldwide.

## 2. Geo-Morphological and Hydrological Settings of Fosso della Chiusa Creek

The Fosso della Chiusa creek (hereafter FdC) has a drainage basin of about 0.92 km^2^, flows above the Ligurian Domain (Jurassic to Cretaceous/Paleocene), and is mostly composed of siliciclastic turbiditic and calcareous-dominant flysch sequences (Figure 1 and Figure 2) [20]. It originates just below the urban area of Abbadia San Salvatore, on the eastern flank of Mt. Amiata, at an altitude of 790 m a.s.l. from the Galleria Italia waters. A few meters from the emergence, a pipeline system diverts about a half of the discharging water to supply a local hydroelectric plant (called “La Turbina” [18]), which is located at the end of the creek course. The other half feeds the FdC creek. Both water streams enter the Pagliola creek, whose waters end in the Paglia River, which is one of the most important tributaries of the Tiber River, the latter crossing Rome before flowing into the Tyrrhenian Sea. According to [21], the anomalous concentrations of Hg measured in the sediments, soils, and waters of about 200 km of the Tiber River Basin, although increasingly diluted, are due to the past mining activity at Mt. Amiata.

The FdC creek has an intermittent regime with an average slope of about 13%, while the width of the riverbed varies from 1 to about 12 m, before entering the Pagliola creek, whose slope is about 3.4% and riverbed width is up to 50 m. The waters discharging from Galleria Italia show a whitish-reddish color (Figure 2a,b), indicative of the presence of suspended Al-Fe-hydroxides. Similarly, the sediments are also dominated by a reddish color [18]. The waters become slightly more transparent about 1 km from the spring, although both sediments and waters still maintain a whitish-reddish color (Figure 2c,d) up to the confluence with the Pagliola creek. The FdC waters are then diluted by Pagliola waters, which are colorless.

## 3. Materials and Methods

The fieldwork was carried out in three sampling surveys (from 4–23 June 2020). FdC water was collected from thirteen sampling sites, about 150–200 m from each other (Figure 1), and labelled GIT0 (at the Galleria Italia emergence) to GIT12 (before the confluence with the Pagliola creek). During the second and third sampling surveys, samples GIT0, GIT3, GIT5, and GIT12 were collected again (Table 1) to verify whether the chemical composition along the FdC creek changed due to a relatively strong rain event which occurred a few days before the second sampling collection (12 June 2020).

The Pagliola creek (PA01) was sampled upstream from the confluence with the FdC creek. From each site, about 1.5 kg of stream sediments were collected. It should be noted that, due to torrential features, at GIT7 the sediment was too tiny to be collected. A reddish colored sediment (Pre-GIT10), deposited from a small cold spring about 50 m to the east of the FdC creek, with a flow rate of <0.3 L/min, was also sampled. Finally, Total Suspended Solids (hereafter TSS) were collected from every two sampling sites: GIT0, GIT3, GIT5, GIT9, and GIT12.

### 3.1. Sampling Procedures

The physicochemical data (pH, T and Eh) were measured with a Crison MM40+ multimeter probe at each sampling site, and different aliquots, stored in polyethylene (PE) bottles, were collected as follows: (a) a filtered (0.45 µm) sample (125 mL) for the determination of the main anionic species and NH_4_^+^ levels; (b) a filtered (0.45 µm) and acidified (0.5 mL of HCl Suprapur, Merck^®^, Darmstadt, Germany) sample (50 mL) for the determination of the main cationic species and SiO_2_ levels; (c) a filtered (0.45 µm) and acidified (0.5 mL of HNO_3_ Suprapur, Merck^®^) sample (50 mL) for the determination of As and Sb levels. A further filtered (0.45 µm) and acidified (0.5 mL with HCl Ultrapur, Romil^®^, Cambridge, UK) sample (50 mL) for the determination of Hg levels was transferred into a 75 mL glass bottle, which was previously cleaned with MilliQ water and diluted Suprapur HCl (Merck^®^) and rinsed at least three times with MilliQ water.

Selected water samples were collected for oxygen and hydrogen isotopes (expressed as δ^18^O and δ^2^H‰ vs. V-SMOW, respectively) and sulfur isotopes of sulfate (expressed as δ^34^S‰ vs. V-CDT) in 125 mL dark glass bottles and 500 mL polyethylene bottles (after filtration at 0.45 µm), respectively. At Galleria Italia (GIT0), and before the confluence with the Pagliola River (GIT12), a hydrometric reel, model ME 4001 SIAP, was used to measure the flow rate. Moreover, an aliquot of 2 L was collected in PE bottles and filtered (in situ) at 0.45 μm for the determination of TSS. The stream sediments were collected using a plastic scoop, which was cleaned with MilliQ water before each sampling, and transferred into plastic bags.

### 3.2. Analytical Methods

**Waters**. Main cations (Na^+^, K^+^, Ca^2+^, and Mg^2+^) and main (Cl^−^ and SO_4_^2−^) and minor (NO_3_^−^ and F^−^) anions were determined at the Department of Earth Sciences (University of Florence) by ion chromatography with 861 Advanced Compact IC-Metrohm and 761 Compact IC-Metrohm, respectively. Bicarbonate (HCO_3_^−^) and NH_4_^+^ were analyzed, within 24 h of sampling, by acidimetric titration (0.01 M HCl and methyl-orange as indicator) using a Multi Dosimat 645-Metrohm and colorimetry, according to the Nessler method, by using a HACH DR2000 molecular spectrophotometer, respectively. As, Sb, and Hg were analyzed by ICP-MS (Agilent 7500CE, detection limit of 0.5 mg/L, 5 μg/L, and 0.1 μg/L for the three chalcophile elements, respectively) at the Accredia Laboratories of CSA Research Group of Rimini, Italy, according to the EPA 6020B 2014 and EPA 7473 2007 methods, respectively. Three replicates were carried out for each sample. The analytical error was <10% [18]. The **δ**^18^O and **δ**^2^H values were determined by CRDS (Cavity Ring-Down Spectroscopy) with a Picarro L2140-i at INGV (Naples, Italy) (analytical error: **δ**^18^O ± 0.08‰ and **δ**^2^H ± 0.5‰), while sulfur isotopes in sulfate were analyzed by EA-IRMS at Iso-Analytical Ltd. (Crewe, UK; analytical error: **δ**
^34^S ± 0.4‰) as described in [22].

**Stream sediments (StS) and Total Suspended Solids (TSS).** The stream sediments, consisting of very fine particles (<180 mm), were dried at 30 °C. Similarly, the 0.45 μm filters of TSS were also dried at 30 °C. Both StS and TSS samples were gently ground in an agate mortar before chemical analyses. The <2 mm StS and <0.45 μm TSS were analyzed for As (EPA 3051A 2007 + EPA 6010D 2018), Sb (EPA 3051A 2007 + EPA 6010D 20189), and Hg (EPA 7473 2007) at the CSA Research Group Laboratories. The analytical error was 10% or lower. Both StS and TSS were analyzed by XRD at the Department of Earth Sciences (University of Florence) to obtain a qualitative determination of the mineral phases by using a Philips PW 1050/37 diffractometer.

## 4. Results

### 4.1. Water Chemical and Isotopic Composition

Temperature (in °C), pH, Eh (in mV), the concentration of main anions and cations and TDS (in mg/L), and the sum of cations and anions (in meq/L) in FdC waters, along with the electroneutrality parameter (Err.%) [23], are reported in Table 1. Temperature varied between 14 and 17.4 °C along the creek. The pH at GIT0 showed a relatively wide interval and varied between 5.17 (23 June 2020) and 7.63 (12 June 2020). The latter was measured a few days after a relatively intense rain event, during which the flow rate increased by up to 70 L/s, with respect to the average yearly flow rate (40 L/s [18]). Along the creek, pH values were mostly slightly alkaline (Table 1). The Eh and TDS values ranged from 32 mV (GIT0, 12 June 2020) to 280 mV (GIT5, 4 June 2020), and from 557 (GIT11, 4 June 2020) to 994 mg/L (GIT1, 4 June 2020), respectively.

Independent of the sampling period, FdC water was Ca^2+^-SO_4_^2−^ in composition (Figure 3), which is the typical composition of surface and ground waters in this area (e.g., [14,18]), while that of PA01 was Ca^2+^-HCO_3_^−^ (SO_4_^2−^). The dominant cation (Ca^2+^) and anion (SO_4_^2−^) showed values of up to 227 and 582 mg/L (GIT1), respectively, and the Ca^2+^/SO_4_^2−^ ratio remained relatively constant over the three sampling sessions (Figure 3). The concentrations of HCO_3_^−^, Cl^−^, Mg^2+^, Na^+^, and K^+^ in the FdC creek were 171, 70, 30, 19, and 15 mg/L, respectively. Except for HCO_3_^−^, the concentrations of main and minor ions in the Pagliola river were lower than those recorded in the FdC creek. Ammonium showed the highest concentrations at GIT0 (4 June 2020), and significantly decreased along the creek. Fluoride and NO_3_^−^ never exceeded 0.18 and 4.1 mg/L, respectively.

Dissolved concentrations of As, Sb, and Hg in FdC and Pagliola waters are reported in Table 2. It should be noted that, in most cases, Sb showed concentrations below or close to the instrumental detection limit (0.1 µg/L), although a slightly higher content was recorded in the Pagliola creek (0.4 µg/L). Arsenic was always >10 µg/L at GIT0, and decreased by an order of magnitude along the FdC creek. Conversely, Hg never exceeded 0.5 µg/L at the emergence, and significantly increased along the flow path up to GIT6 (up to 2.8 µg/L on 4 June 2020). A slight decrease was observed in the lower reaches of the FdC creek. In the Pagliola creek (before the confluence with the FdC creek), Hg was 0.3 µg/L.

The water and sulfur isotope values are reported in Table 2. The **δ**^18^O and **δ**^2^H values ranged from −7.8‰ (GIT0) to −6.4‰ (GIT12, 5 June 2020) vs. SMOW, and from −48.6‰ (GIT0, 4 June 2020) to −35.8‰ (GIT12, 5 June 2020) vs. SMOW, while those of Pagliola creek were −5.5 and −31.9‰ vs. SMOW, respectively. The **δ**^34^S-SO_4_ values were rather homogeneous along the FdC creek, ranging between −4.6‰ (GIT9) and −4.0‰ (GIT0) vs. V-CDT. A significantly more negative value was measured in the Pagliola creek (−10‰ vs. V-CDT).

### 4.2. StS and TSS

**StS.** Similar to what was recorded from the water samples, the content of Sb in StS (as well as in TSS; see below) was equal to or below the detection limit (<1 mg/kg). Such low Sb concentrations do not provide specific information about its behavior in the FdC matrices; consequently, antimony does not pose any criticalities and it will not be further discussed. The concentrations of As along the FdC creek varied between 336 (GIT0) and 7 mg/kg (GIT12), and showed a general decrease from the source to the confluence with the Pagliola creek. It should be noted that the sediment collected near the small spring (Pre-GIT10) close to the main FdC course (Figure 1) showed a concentration of As of 137 mg/kg. The lowest As concentrations were measured in the Pagliola creek sediment (2 mg/kg). Different behavior was observed for Hg, as the lowest concentrations were recorded at GIT0 (12.5 mg/kg), GIT5 (7.5 mg/kg), GIT7 (8 mg/kg), and Pre-GIT10 (7 mg/kg), while the highest concentrations were measured in the central and lower reaches of the FdC creek (GIT12 = 153 mg/kg) (Table 2). The sediment collected from the Pagliola creek had a Hg concentration of 0.8 mg/kg.

The qualitative XRD analysis allowed recognition of the presence of kaolinite all along the FdC creek, while quartz, feldspar, and calcite were found from GIT6 down to the confluence, as well as in PA01. GIT0 was mostly dominated by goethite, likely related to the relatively high content of Fe measured in the GIT0 waters [15]. The pre-GIT10 sediment was characterized by goethite, illite, muscovite and quartz.

**TSS.** As previously mentioned, the concentration of Sb in TSS was equal to or <1 mg/kg, while those of As and Hg ranged from 41 (GIT9) to 77 (GIT0) mg/kg and from 0.27 (GIT0) to 17.5 (GIT9) mg/kg, respectively. Similar to the observations of the water samples, it is remarkable that the concentration of As, though to a minor extent, decreased from the source to the confluence, whereas that of Hg increased up to GIT10 (17.5 mg/kg) and registered an abrupt decrease at GIT12 (4.71 mg/kg). Kaolinite and quartz were found in all TSS samples.

## 5. Discussion

### 5.1. Water Characterization and Isotopic Fingerprint Using δ^18^O, δD and δ^34^S

Similar to other surface and ground waters analyzed within the former mining area of Abbadia San Salvatore and its surroundings, the FdC creek water chemistry is Ca(Mg)-SO_4_ [14,15,18], and distinctly differs from that of the Pagliola creek (Ca(Mg)-HCO_3_). The FdC waters resemble, in terms of major and minor dissolved species, those of Galleria Italia (GIT0) [18]. This suggests that no significant inputs from other surface or spring waters, with different chemical compositions, occur along the FdC creek path. This is also supported by the chemical composition determined during the three surveys carried out during this work (Table 1). A dilution process due to meteoric waters is, apparently, the sole process affecting the FdC creek water, also supported by the Ca^2+^/SO_4_^2+^ ratio, which remains relatively constant independent of the sampling period.

According to [18], the chemistry of the FdC creek circumneutral waters is related to water–rock interactions processes involving relatively soluble minerals, mostly sulfates and, subordinately, carbonates, at which point a silicate component is added. About 80% of the TDS value is related to the sum of Ca(Mg) and SO_4_, whereas HCO_3_ and Na + Cl are always clustered around 100 and <35 mg/L, respectively; i.e., <20% of the total TDS. In contrast, in the Pagliola creek (PA01) the Ca-HCO_3_ component largely dominates, due to dissolution processes mostly affecting carbonate-rich rocks.

By plotting the δD vs. δ^18^O‰ V-SMOW values for each sampling site in the binary diagram of Figure 4, the FdC creek water samples were shown to be aligned along a meteoric line (δD = 7.09 ∗ δ^18^O + 6.8), which is roughly parallel to the global (GMWL; [24]) and Central Italy meteoric water lines (CIMW, [25]), suggesting, as expected, their meteoric origin. As reported by [18], during mining activity hot water was found to mix with the water draining the galleries. This implies that either the thermal waters are meteoric-derived, or possible other contributions are masked by the rain waters feeding Galleria Italia.

It is noteworthy to point out that water isotopes tend to become heavier as they travel from the emergence (at about 790 m) to the confluence with the Pagliola River (at about 480 m), the latter being characterized by more positive δD and δ^18^O values. Setting aside the water isotopes of GIT12, sampled during the three sampling campaigns, which tend to progressively approach that of PA01, the distribution of FdC waters indicates that no significant evaporation processes affect the water samples (Figure 4), likely as a result of the torrential character of the creek. Altitude plays the most important role, as supported by the altitude (in m) vs. δ^18^O binary diagram (Figure 4b), which shows that, at a relatively steep decrease in terms of elevation, oxygen isotopic values are concentrated in a narrow, decreasing range. The isotopically heavier samples are those originating before the confluence with the Pagliola River, where the FdC slope is lower than the upper reaches of the creek. Assuming an isotopic gradient of −0.2‰/100 m, which characterizes the rain waters from the western part of Italy, e.g., [25], the waters discharging from Galleria Italia (GIT0) enter the shallow underground circuit at an elevation of about 1100 m.

To better define the source of sulfur, considering that the study site was located in a former Hg-mining area where cinnabar was exploited and processed to produce liquid mercury, the δ^34^S-SO_4_ values were measured. The isotopic results are scarcely variable: from −4.6 to −4.0‰ V-CTD, except for PA01, where a significant negative value was measured (−10.5‰ V-CTD). To the best of our knowledge, the only sulfur isotopic value in dissolved sulfate from the waters of Galleria Italia (i.e., GIT0) showed a value of −4.9‰ [15], which is in agreement with those measured in the present work. According to [26,27], the δ^34^S values of sulfur isotopes from cinnabar are rather variable, ranging between −7 and +2‰ V-CTD, with most samples clustering around −1 ± 3.2‰ V-CTD. The levels of FdC sulfur isotopes in sulfate approach those recorded in the shallow groundwater system occurring inside the mining area of Abbadia San Salvatore [14], which is located at about 500 m to the north of FdC. The more negative values could be related to: (i) dissolution and oxidation processes of reduced S-bearing minerals; (ii) SO_4_-bearing evaporites; (iii) alteration of waste material derived by roasting processes the meteoric waters are interacting with; or (iv) a combination of different sources. However, we cannot exclude the oxidation of dissolved H_2_S along the FdC path, as also shown by increasing redox values (Table 1). Nevertheless, further investigation is required to better define the source of SO_4_.

### 5.2. Geoaccumulation Index (Igeo)

The geoaccumulation index I_geo_ was first introduced by [28] to assess metal contamination in sediments by considering current concentrations vs. those of the pre-industrial era. In this paper, I_geo_ was only applied to As and Hg since, as previously mentioned, Sb concentrations were, in most cases, approaching or below the instrumental detection limit. The following equation was used:I_geo_ = log_2_ [C_n_/(B_n_ ∗ 1.5)]
where C_n_ is the concentration of the element in the fine fraction of the studied sediments, and B_n_ is the geochemical background value referring to clay rocks or stream sediments. The latter can be either measured or inferred from the literature. In this work, the adopted background values of As and Hg were 7 [29] and 0.3 [30,31] mg/kg, respectively, which refer to stream sediments from southern Tuscany in the surroundings of Mt. Amiata. According to the I_geo_ value, each sediment was then classified in 7 different classes, indicating progressive contamination (from 0 to 6 [28]). The computed classes, according to the I_geo_ values, of As and Hg are listed in Table 3.

The spatial distribution of the As and Hg classes of the FdC stream sediments are visualized in Figure 5, where each class is distinguished by different colors: from white (class 0) to red (class 6), with the exception of GIT3 (in blue), which was not classified. The stream sediment from GIT0 was classified as extremely contaminated (class 6), whilst those from GIT1 to GIT7 were assigned to class 2 and 3, respectively. Eventually, the stream sediments from GIT9 to GIT12 and PA01 become classified as class 0 (uncontaminated; Figure 5a), suggesting that As is preferentially partitioned in the solid phase close to the source (GIT0).

The I_geo_ classes of Hg (Figure 5b) were differently distributed with respect to those of As, since classes 5 and 6 largely dominated the FdC stream sediments. The PA01 stream sediments, before the confluence with the FdC creek, pertains to Class 1 for Hg (uncontaminated to moderately contaminated; Table 3). Despite the fact that no I_geo_ values for the stream sediments of the Pagliola creek after the confluence of FdC were available, [32] reported that the Hg concentrations downstream the confluence varied between 5 and 53 mg/kg. Consequently, as far as the concentrations of As and Hg in the stream sediments are concerned, and in agreement with [33], it is possible to conclude that, while concentrations of of As tend to abruptly decrease a relatively short distance from the source, those of Hg are more environmentally impactful as their effects are able to reach medium-to-large distances.

The relatively high Hg concentrations measured in the FdC stream sediments (up to 153 mg/kg) are comparable with those found in the Madeira and Elbe rivers [34,35]. Much higher concentrations (from 100 to 1000 mg/kg) were recorded in the stream sediments analyzed by [36] from the former Hg-mining area of Idrija (Slovenia), and attributed to the mining processes during the exploitation of cinnabar. Concentrations up to 6000 mg/kg were also reported by [36] and reference therein in the stream sediments from the surroundings of Suplja Stena (Mt. Avala, near Belgrade, Serbia). In the central Balkan peninsula, high As concentrations were also measured (up to 10 mg/kg; [37]) and associated with active mines and metallurgical and electric-power plants. Similar to what was recorded in the FdC creek, the As concentrations in the stream sediments of Foster creek (Bonanza mine, Oregon) varied from 21 to 109 mg/kg [38,39]) and a decrease in concentration, similar to levels recorded in the FdC creek, from the source to the confluence with the main riverine system (Calapooya creeks) was observed [38].

### 5.3. Partition Coefficients (log-Kd) and Yearly Estimate Mass Load of As and Hg

To better characterize the fate and behavior of As and Hg in the aquatic system of FdC, the log-transformed partition coefficients (Kd), of the sampling sites where the TSS concentrations of As and Hg were available (GIT0, GIT3, GIT5, GIT9, and GIT12), were calculated, as follows:Log-K_d_(L/kg) = [TSS_M__e_(mg/kg)]/[T_Me_(μg/L)/1000]
where TSS_Me_ is the concentration of As or Hg and T_Me_ is defined as the Total Metal Concentration for As or Hg in the liquid phase, after filtration at 0.45 μm [40]. The log-transformed partition coefficients (log-Kd) for FdC are shown in Table 4. The log-K_d_ values of Hg range from 2.83 (GIT0, 4 June 2020) to 3.56 (GIT12, 5 June 2020, before the confluence the Pagliola creek), while those of As vary between 3.81 (GIT0, 4 June 2020) and 4.94 (GIT12 of 5 June 2020), suggesting that both elements are preferentially partitioned in the suspended phase (TSS). However, the As Log-K_d_ values are significantly higher than those computed for Hg, indicating that arsenic has a higher affinity to TSS than to water, the former being likely characterized by Fe-hydroxides onto which As can be absorbed, e.g., [41,42,43,44]. This is further supported when the log-K_d_ values of As and Hg between the stream sediments and TSS are considered, since the former tend to be <1, whereas those of Hg are up to 2. This would explain why the concentrations of Hg are much higher in stream sediments, with respect to As. However, geomorphological characteristics may play a critical role in the partitioning of the two elements, since, according to [33], slope is likely a driving factor in the deposition of suspended solids from riverine systems. The morphology of the first sector of the FdC creek is indeed relatively steep, while the central part is smoother, thus favoring the deposition and adsorption of As and Hg. No specific investigations were carried out in this respect, since this falls outside the scope of this work.

### 5.4. Yearly Estimate Mass Load of As and Hg

The evaluation of the amount of dissolved and transported As and Hg from waters associated with active and abandoned mines is a critical environmental issue (e.g., [33,36]). In order to compute the amount of the two elements discharged from the FdC creek into the Pagliola creek, the Total Mass loads of As and Hg, before the confluence (GIT12), were considered. The mean flow rate (Q) of GIT0 was assumed to be of 40 L/s, which represents the yearly average value. Similarly, the concentrations of T_Me_ and TSS_Me_ were considered constant (Table 2). Thus, the Total Mass load was computed, as follows:Total Mass load (kg/year) = {[TSS_Me_(g/L) + T_Me_(g/L)] ∗ Q(L/s)}
where T_Me_ is defined as the total metal concentration (As or Hg) in the filtered (at 0.45 μm) liquid aliquot and TSS_Me_ is the total suspended solid fraction >0.45 μm (g/L). As previously mentioned, the mean flow rate of the discharging water from Galleria Italia (GIT0) is 40 L/s, which is the amount of water measured at GIT0 in June 2020; 50% of this flow feeds the FdC creek, whilst the remaining water is diverted to the hydroelectric plant. This agrees with the flow rate of the FdC creek measured before the confluence with the Pagliola creek during our survey (21 L/s at GIT12). This implies that water inputs along the FdC creek can be considered negligible. The computed loads were about 1.3 kg/year and 0.7 kg/year of Hg and As, respectively. If the partitioning coefficient between the dissolved and suspended discharge is considered, Hg and As mostly enter the Pagliola creek as <0.45 µm fractions (99.6 and 91.1%, respectively), whereas the suspended Hg and As loads correspond to 0.4 and 8.9%, respectively. A computed value of 11 kg year^−1^ was reported by [33], which represents the estimated mass load of Hg (Table 4) discharging from the Paglia river, whose main tributary is the Pagliola creek, into the Tiber River. This means that the contribution of Hg from the FdC creek is about 1/10.

By comparing the yearly mass load of Hg from the FdC creek with other mining-affected areas, such as the Isonzo River (ca. 1500 kg year^−1^, [45]), which drains the entire Idrija area, and the Guadalupe River, which drains New Almadén, California [46], it is possible to state that the amount of Hg discharged by FdC is relatively low (Table 5). However, if we consider the specific yearly Hg mass load, i.e., the total mass load divided by the drainage basin (see [40]; Table 5), the FdC creek can be regarded as a significant source of Hg (Table 5, Figure 6). Basically, the FdC yearly mass load of Hg is comparable with those of the Bonanza mine in western Oregon (Table 5, 0.75 kg year^−1^; [47]) and the San Carlos creek, which drains the New Idrija Hg mine, California (1.5 kg year^−1^; [48]).

## 6. Conclusions

The circumneutral Ca(Mg)-SO_4_ waters that discharge at Galleria Italia and feed the Fosso della Chiusa Creek originate from meteoric waters that enter the mining galleries of the former Hg-mining area of Abbadia San Salvatore (Tuscany, central Italy) at an elevation of about 1100 m. They are characterized by relatively high concentrations of As (up to 12 mg/L) and Hg (up to 2.8 mg/L). The stream sediments and total suspended solids also show high As and Hg concentrations, up to 336 and 77 mg/L and 153 and 17.5 mg/L, respectively. Differences, in terms of geochemical behavior, are seen between As and Hg when their distribution in water, total suspended solids, and stream sediments is considered. The log-Kd values showed evidence that Hg is preferably partitioned in the solid phase. Arsenic levels peak in stream sediments just after emergence from Galleria Italia, downstream of which it tends to be adsorbed in the suspended solid. Consequently, from an environmental point of view, FdC stream sediments are most importantly impacted by Hg with respect to As; I_geo_ Hg-classes were classified as heavily to extremely contaminated. The total mass loads of As and Hg released from FdC are relatively low, although, when computing the specific mass load, values of Hg are comparable to, or even higher than, those reported in the literature for rivers interacting with Hg-mines or -ore deposits.

According to our results, the impact by Hg and, though at a minor extent, As on the riverine system can be minimized by creating various pools along the water course to favor the deposition and precipitation of Hg- and As-rich sediments. However, the deposited sediments in the pools should periodically be removed and treated or disposed in specific landfills, to avoid that intense precipitation would pour the accumulated material into the Pagliola Creek. An economic evaluation and a risk assessment must be considered before undertaking any action.

## Figures and Tables

**Figure 1 ijerph-19-05131-f001:**
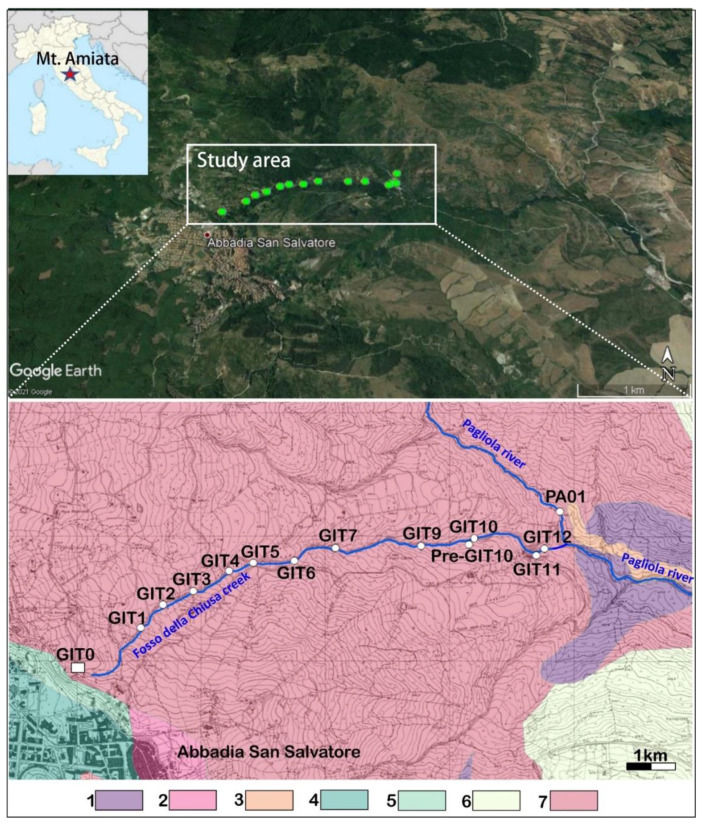
Location of the Mt. Amiata area and the Fosso della Chiusa creek (above) along with the schematic geological map (below) (modified from the Geological Database of Regione Toscana: http://www502.regione.toscana.it/geoscopio/geologia.html#, accessed on 4 June 2021, in which the IDs of the sampling sites are reported. 1, landslide; 2, detritus; 3, recent and present alluvial deposits (Holocene); 4, eluvial and colluvial deposits (Holocene); 5, rhyo-dacitic volcanics (Pleistocene); 6, (basal olistostrome) dark-greyish clays with blocks of marly limestone, calcareous microbreccia, and sandstone (Upper Oligocene); 7, Ligurian domain: clays and marls with limestone, marly limestones, sandstone, and calcarenite, occasionally intercalated with sedimentary breccia consisting of green rocks, jaspers, and limestones (Paleocene-Lower Cretaceous).

**Figure 2 ijerph-19-05131-f002:**
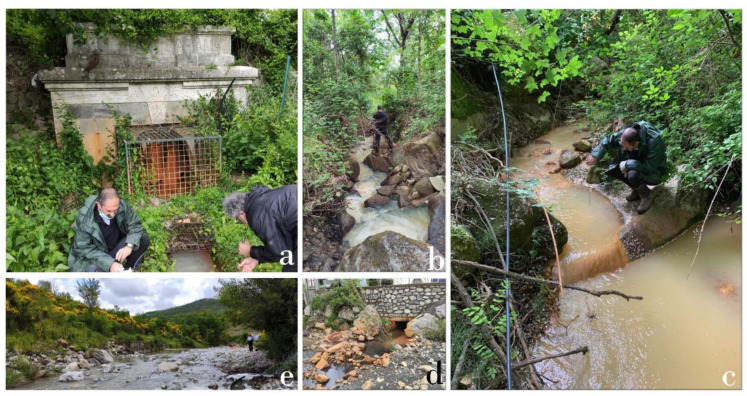
(**a**) Waters discharging from Galleria Italia (GIT0) feeding the Fosso della Chiusa creek. Note the whitish-reddish waters at the GIT0, GIT3 (**b**), a couple of hundreds meter from the emergence, and GIT6 (**c**), about 1 km from the emergence, sampling sites; (**d**) FdC waters entering the Pagliola creek after supplying the “La Turbina” hydroelectric plant; (**e**) the Pagliola creek after the confluence with FdC.

**Figure 3 ijerph-19-05131-f003:**
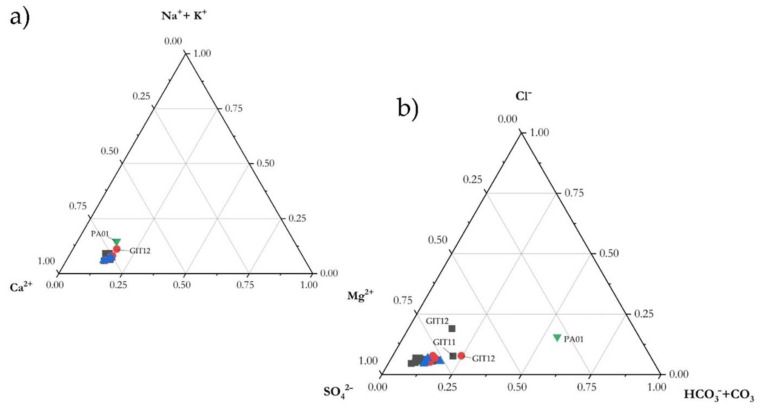
Cationic (**a**) and anionic (**b**) triangular diagrams of the studied waters (FdC sampling: 4 June 2020 black square; 12 June 2020 blue triangle; 23 June 2020. Pagliola creek: green triangle).

**Figure 4 ijerph-19-05131-f004:**
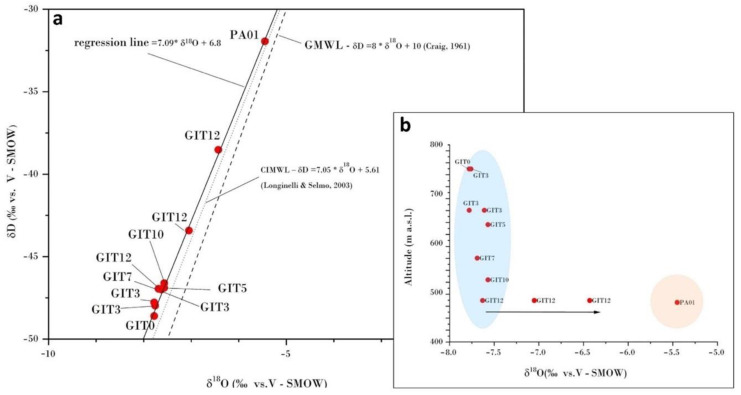
(**a**) The binary diagram of δD vs. δ^18^O in FdC waters (red circles). The Central Italy (CIMWL, [25]), and the Global Meteoric Water (GMWL, [24]) lines are reported for comparison with the regression line depicted by the studied waters. Graph (**b**) correlates the altitude (in m) vs. the δ^18^O values. The blue field contains most of the FdC waters, whose isotopic oxygen values decrease with altitude. The black arrow points to the isotopic composition of the FdC waters collected before entering the Pagliola river and the Pagliola creek itself.

**Figure 5 ijerph-19-05131-f005:**
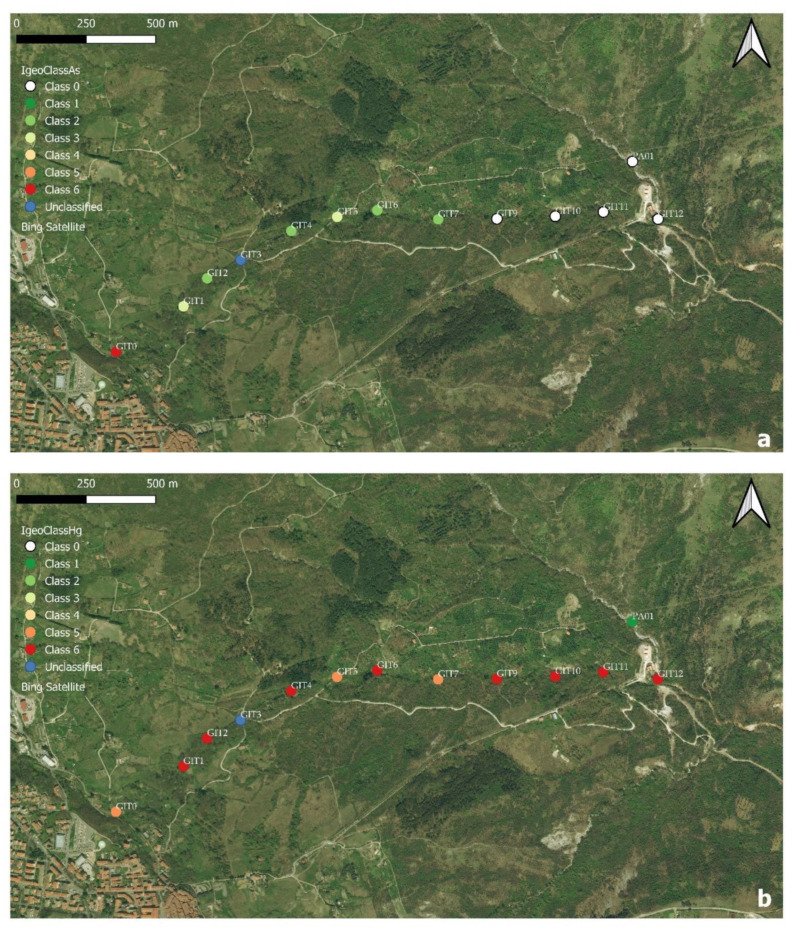
Sampling sites and I_geo_ classes [19] along FdC: (**a**) Arsenic and (**b**) Mercury. The easternmost portion of Abbadia San Salvatore and the location of PA01, corresponding to the Pagliola creek before the confluence of the FdC creek, are visible in the lower left and upper right corners, respectively. The two maps were created with QGIS using Bing Satellite as the base-layer. Legend (according to [19,28]): Class 0 = “uncontaminated”; Class 1 = “from uncontaminated to moderately contaminated”; Class 2 = “moderately contaminated”; Class 3 = “from moderately contaminated to heavily contaminated”; Class 4 = “heavily contaminated”; Class 5 = “from heavily to extremely contaminated”; Class 6 = “Extremely contaminated”. Sample GIT3 was reported in blue, since no data were available.

**Figure 6 ijerph-19-05131-f006:**
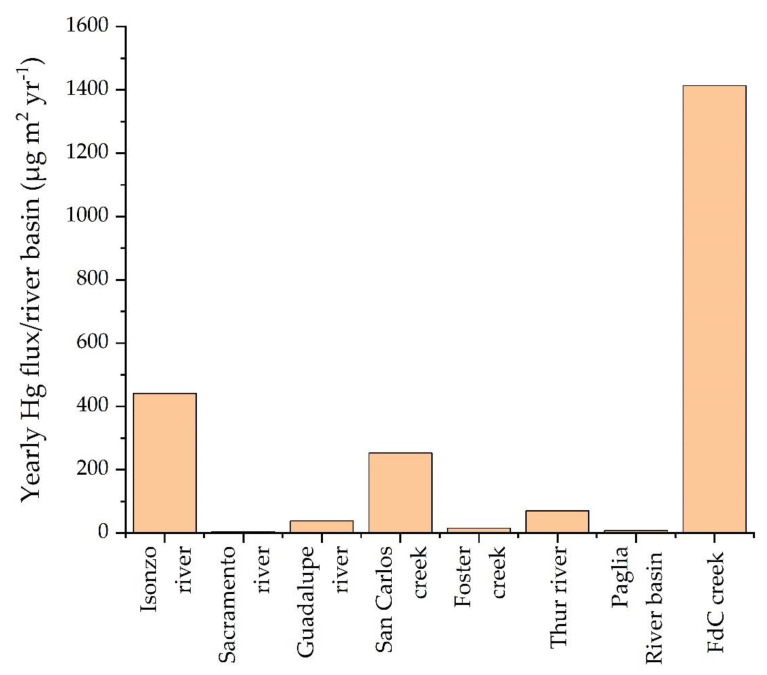
Bar diagram of the specific mass load from different active and abandoned mining localities in comparison with that computed for FdC.

**Table 1 ijerph-19-05131-t001:** Physicochemical data and concentrations of main and minor anions, cations, and TDS (in mg/L), measured along the Fosso della Chiusa creek and in the Pagliola creek before the confluence of the FdC creek. Scat and San are in meq/L, while Err. (in %) is expressed as ((Scat − San)/(Scat + San)) ∗ 100.

Sample ID	Sampling Date	Distance from Galleria Italia	T	pH	Eh	HCO_3_^−^	F^−^	Cl^−^	NO_3_^−^	SO₄²⁻	Ca^2+^	Mg^2+^	Na^+^	K^+^	NH_4_^+^	Scat	San	Err.	TDS
		Km	(°C)		mV	(mg/L)	(mg/L)	(mg/L)	(mg/L)	(mg/L)	(mg/L)	(mg/L)	(mg/L)	(mg/L)	(mg/L)	(meq/L)	(meq/L)	%	(mg/L)
GIT0	4 June 2020	0.0	14.3	6.01	55.0	103	0.18	29	0.5	511	195	22	11	9.6	3.38	12.48	13.16	−2.64	885
GIT0	12 June 2020	0.0	15.0	7.63	32.0	104	0.03	21	0.4	446	198	24	12	9.7	2.64	12.77	11.59	4.84	818
GIT0	23 June 2020	0.0	14.7	5.17	83.0	108	0.07	22	1.7	547	205	23	12	10.5	2.31	13.01	13.81	−2.97	931
GIT1	4 June 2020	0.3	14.7	7.44	126.0	70	0.01	23	1.5	582	227	30	16	14.7	0.32	14.89	13.93	3.32	964
GIT2	4 June 2020	0.4	14.8	7.94	195.0	70	0.07	28	2.7	449	184	25	12	10.9	0.32	12.03	11.32	3.03	781
GIT3	4 June 2020	0.6	15.0	7.63	182.0	70	0.06	32	4.1	531	186	24	14	10.4	0.22	12.14	13.17	−4.05	871
GIT3	12 June 2020	0.6	14.0	7.03	204.0	93	<0.01	29	1.5	395	173	24	13	15.4	0.09	11.56	10.60	4.32	744
GIT3	23 June 2020	0.6	16.0	7.59	122.0	92	0.02	27	3.8	441	179	23	12	9.5	0.02	11.57	11.52	0.18	788
GIT4	4 June 2020	0.7	15.0	6.86	114.0	73	<0.01	22	1.7	454	187	24	15	10.3	0.14	12.21	11.32	3.77	788
GIT5	4 June 2020	0.8	15.0	7.79	280.0	73	0.04	22	1.8	521	183	23	14	10.1	0.02	11.88	12.72	−3.43	849
GIT5	12 June 2020	0.8	14.0	7.76	57.0	106	0.04	27	1.3	415	168	22	13	12.1	0.09	11.09	11.17	−0.33	765
GIT5	23 June 2020	0.8	15.2	7.82	118.0	99	0.02	24	1.2	497	179	24	13	11.4	0.08	11.78	12.66	−3.59	849
GIT6	4 June 2020	1.0	15.0	7.75	189.0	72	0.07	24	2.6	469	180	24	15	11.4	0.01	11.94	11.67	1.14	799
GIT7	4 June 2020	1.3	15.0	6.86	236.0	71	0.04	22	2.3	406	170	23	12	7.6	0.01	11.08	10.26	3.87	713
GIT9	23 June 2020	1.5	16.8	7.60	141.0	111	<0.01	23	1.6	450	194	25	15	8.3	0.02	12.56	11.89	2.77	829
GIT10	23 June 2020	1.7	16.0	7.57	143.0	124	0.02	26	2.4	444	203	25	13	9.6	0.06	13.01	12.05	3.82	848
GIT11	4 June 2020	1.9	15.0	7.75	122.0	103	0.01	21	2.4	264	132	14	14	7.2	0.01	8.56	7.81	4.60	557
GIT12	4 June 2020	2.0	15.0	8.26	102.0	99	<0.01	70	2.9	326	144	18	14	10.3	0.03	9.51	10.44	−4.68	684
GIT12	12 June 2020	2.0	14.0	7.12	223.0	161	0.06	30	1.6	349	154	23	19	14.7	0.08	10.75	10.77	−0.07	752
GIT12	23 June 2020	2.0	17.4	7.60	122.0	132	0.01	23	1.0	435	173	23	12	10.4	0.10	11.35	11.90	−2.37	811
PA01	4 June 2020	2.2	15.0	7.89	180.0	179	0.02	29	1.7	75	78	10	17	2.4	0.05	5.57	5.37	1.86	394

**Table 2 ijerph-19-05131-t002:** Oxygen and hydrogen isotopes in water (expressed as δ^18^O and δD vs. ‰ V-SMOW, respectively) as well as sulfur isotopes (in SO_4_^2−^; expressed as δ^34^S vs. ‰ VCDT) and As, Sb, and Hg concentrations in water (in µg/L), selected suspended solids, and stream sediments (in mg/kg) from the FdC and Pagliola creeks.

		Liquid Phase						Suspended Solids			Sediments		
Sample ID	Date	δ^18^O	δ^2^H	δ^34^S-SO_4_	As	Sb	Hg	As	Sb	Hg	As	Sb	Hg
		‰ vs. SMOW	‰ vs. SMOW	‰ vs. CDT	µg/L	µg/L	µg/L	mg/kg	mg/kg	mg/kg	mg/kg	mg/kg	mg/kg
GIT0	4 June 2020	−7.8	−48.6	−4.5	12.0	<0.1	0.4	77	< 1	0.27	336	<1	12.5
GIT0	12 June 2020	−7.8	−48.0	−4.0	10.9	<0.1	0.5						
GIT0	23 June 2020	−7.8	−47.8		10.9	<0.1	0.1						
GIT1	4 June 2020				2.0	<0.1	1.6				43	<1	39.3
GIT2	4 June 2020				1.2	<0.1	1.4				36	<1	68.0
GIT3	4 June 2020	−7.6	−46.9	−4.4	1.0	<0.1	1.8	62	<1	1.8			
GIT3	12 June 2020				1.0	<0.1	1.9						
GIT3	23 June 2020				1.0	<0.1	1.6						
GIT4	4 June 2020				1.0	<0.1	2.2				21	<1	105.0
GIT5	4 June 2020	−7.6	−46.9	−4.3	0.9	<0.1	2.1	54	<1	15.3	68	<1	7.5
GIT5	12 June 2020				0.8	<0.1	1.6						
GIT5	23 June 2020	−7.6	−47.0		0.9	<0.1	1.3						
GIT6	4 June 2020				1.0	<0.1	2.8				29	<1	100.0
GIT7	4 June 2020	−7.7	−47.0		1.2	<0.1	2.2				41	<1	8.1
GIT9	23 June 2020	−7.6	−46.6	−4.6	0.7	0.1	<0.1	41	<1	17.5	8	<1	75.0
GIT10	23 June 2020				0.7	0.1	1.4				11	<1	80.0
GIT11	5 June 2020				0.6	0.2	1.8				9	1	53.9
GIT12	5 June 2020	−6.4	−38.5	−4.2	0.6	0.2	1.3	52	<1	4.71	7	<1	153.0
GIT12	12 June 2020	−7.1	−43.4		0.7	0.2	<0.1						
GIT12	23 June 2020	−7.6	−46.9	−4.5	0.6	0.1	<0.1						
PA01	5 June 2020	−5.5	−31.9	−10.5	0.3	0.4	0.3				2	1	0.8
Pre-GIT10 spring	23 June 2020										137	<1	7.0

**Table 3 ijerph-19-05131-t003:** Classes of As and Hg according to the I_geo_ values of the FdC stream sediments. The classes are, according to [28]: Class 0 = “uncontaminated”; Class 1 = “from uncontaminated to moderately contaminated”; Class 2 = “moderately contaminated”; Class 3 = “from moderately contaminated to heavily contaminated”; Class 4 = “heavily contaminated”; Class 5 = “from heavily to extremely contaminated”; Class 6 = “extremely contaminated”.

Sample	Class	As	Class	Hg
GIT0	6	extremely contaminated	5	heavily to extremely contaminated
GIT1	3	moderately to heavily contaminated	6	extremely contaminated
GIT2	2	moderately contaminated	6	extremely contaminated
GIT3	-	-	-	-
GIT4	2	moderately contaminated	6	extremely contaminated
GIT5	3	moderately to heavily contaminated	5	heavily to extremely contaminated
GIT6	2	moderately contaminated	6	extremely contaminated
GIT7	2	moderately contaminated	5	heavily to extremely contaminated
GIT9	0	uncontaminated	6	extremely contaminated
GIT10	0	uncontaminated	6	extremely contaminated
GIT11	0	uncontaminated	6	extremely contaminated
GIT12	0	uncontaminated	6	extremely contaminated
PA01	0	uncontaminated	1	uncontaminated to mod. contaminated

**Table 4 ijerph-19-05131-t004:** Log-transformed partition coefficient (reported as log Kd) for As and Hg along FdC.

Sample ID	Q	log K_d_			Mass Load (kg/year)				
	L/s	As	Hg	DAs	TSSAs	As Total	DHg	TSSHg	Hg Total
GIT0	40	3.81	2.83	13.32	6.80	20.12	0.44	0.04	0.48
GIT3	-	4.79	3.00	-	-	-	-	-	-
GIT5	-	4.77	3.86	-	-	-	-	-	-
GIT9	-	4.77	-	-	-	-	-	-	-
GIT12	40	4.94	3.56	0.67	0.07	0.73	1.33	0.01	1.34
PA01	213	-	-	-	-	-	-	-	-

**Table 5 ijerph-19-05131-t005:** Yearly mass load (kg/year^−1^) of Hg from different rivers affected by Hg-rich ore deposits, along with river basin area and specific mass load (yearly mass load divided by basin area) data.

River Name	Location	Reference	Hg Mass Load	River Basin Area	Yearly Hg Flux/River Basin
			kg/year^−1^	km^2^	μg m^2^ year^−1^
Isonzo River	Idrija (Slovenia)	[41]	1500	3400	441
Sacramento River	California (USA)	[49]	61–470	71432	3.7
Guadalupe River	New Almadèn, California (USA)	[46]	4–30	440	38.6
San Carlos Creek	New Idrija, California (USA)	[48]	1.5	5.94 **	253
Foster Creek	Bonanza mine, Oregon	[47]	0.75	50.4 **	14.9
Thur River	France, Swiss	[40]	34–86 m^2^ year^−1^ *	270	70
Paglia River Basin (PRB)	Tuscany and Lazio, Italy	[33]	11	1330	8.3
Fosso della Chiusa Creek	Tuscany, Italy	this work	1.33	0.92	1413

* low-high water flow period; ** estimated values.

## Data Availability

Not applicable.

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
