# Peer review of "Mercury and Arsenic Discharge from Circumneutral Waters Associated with the Former Mining Area of Abbadia San Salvatore (Tuscany, Central Italy)"

_ijerph, 2022, doi:10.3390/ijerph19095131_

Round 1
Reviewer 1 Report
Mercury and arsenic discharge from circumneutral waters associated with the former mining area of Abbadia San Salvatore (Tuscany, central Italy)
The manuscript Mercury and arsenic discharge from circumneutral waters associated with the former mining area of Abbadia San Salvatore (Tuscany, central Italy) refers to a very important problem, which is the pollution of waters and bottom sediments with toxic elements. The manuscript has a standard structure that includes the elements specified in the manual for journal authors. It is divided into the following sections: Introduction, Materials and Methods, Results, Discussion and Conclusions.
Notes for the manuscript are provided below.
- Delete the period in the title of the manuscript
- The paper does not indicate what research problem the authors want to analyze, there is no precisely presented research hypothesis.
- Are the field studies carried out in the three studies representative and sufficient to determine the water quality? Are these not random results?
- Line 194, (Err.%)?
- Line 199, correct "40 L / sec"
- The Discussion chapter is poorly written. I suggest that it be rewritten. Some of the descriptions should be moved to the Materials and Methods and Results chapter and focus on the discussion of the obtained research results with other studies.
- Conclusions are overly elaborate and vague. They need to be redrafted
- The References chapter contains a large number of literature items, which may indicate a thorough analysis of the research problem, but it should be stated that the authors mainly rely on quite old literature because only about 13% of it is younger than 5 years.
Author Response
Here below the file with the notes for the reviewer #1

Reviewer 2 Report
The manuscript “Mercury and arsenic discharge from circumneutral waters associated with the former mining area of Abbadia San Salvatore 3 (Tuscany, central Italy)” reports on a geochemical study of the watersheds impacted by mining reporting results for several heavy metals, with a focus on Hb and As. The data are interesting and show how concentrations change downstream from the mining area, both in suspended sediments and the dissolved fraction.
While the data are interesting, the paper is difficult to read due to issue with English grammar. While mostly the intent can be inferred, in some cases it is not clear what the authors are trying to convey.
I recommend the authors find a native English speaker that is familiar with geochemistry to help them edit the manuscript.
In general, the paper reports data and concentrations, but does report or speculate on driving forces. We are left to try to infer what various data results mean. I would have preferred more analysis of the results. Often there is other information, available to those familiar with a site, that can help explain results. For example, the fact the Hg and arsenic concentrations behave differently downstream. I would have like some analysis and potential processes or causes. If this research cannot answer that question, that is fine, but more discussion on why it could not be answered and what was explored would be helpful.
General comments
e.g., and i.e., both should have commas at the end.
There is a problem with symbols e.g., L151-152 and many other places
I have provided a markup of the document that highlights some of the English issues, though it is not meant as a comprehensive edit, just marking some examples.
Recommendation
Major rewrite – fix English issues, analyze the data and tell me what it means, don’t just report it.

Author Response
Here below the reply for the reviewer #2

Round 2
Reviewer 1 Report
The authors of the manuscript addressed all the comments in the review.